# Molecular Progression of Myeloproliferative and Myelodysplastic/Myeloproliferative Neoplasms: A Study on Sequential Bone Marrow Biopsies

**DOI:** 10.3390/cancers13225605

**Published:** 2021-11-09

**Authors:** Magdalena M. Brune, Achim Rau, Mathis Overkamp, Tim Flaadt, Irina Bonzheim, Christian M. Schürch, Birgit Federmann, Stefan Dirnhofer, Falko Fend, Alexandar Tzankov

**Affiliations:** 1Institute of Medical Genetics and Pathology, University Hospital Basel, Schönbeinstrasse 40, CH-4031 Basel, Switzerland; magdalena.brune@usb.ch (M.M.B.); stefan.dirnhofer@usb.ch (S.D.); 2Institute of Pathology and Neuropathology, University Hospital Tübingen, 72076 Tübingen, Germany; achim.rau@med.uni-tuebingen.de (A.R.); mathis.overkamp@googlemail.com (M.O.); tim.flaadt@med.uni-tuebingen.de (T.F.); irina.bonzheim@med.uni-tuebingen.de (I.B.); christian.schuerch@med.uni-tuebingen.de (C.M.S.); birgit.federmann@med.uni-tuebingen.de (B.F.); 3Institute of Pathology, University of Bern, Murtenstrasse 8, CH-3008 Bern, Switzerland

**Keywords:** myeloproliferative neoplasm, myelodysplastic/myeloproliferative neoplasm, secondary acute myeloid leukemia, genetic testing, mutation analysis, risk stratification

## Abstract

**Simple Summary:**

Myeloid neoplasms (MN) are malignant hematopoietic stem cell disorders, which can progress into aggressive forms of blood cancer, likely due to the acquisition of additional genetic alterations. We investigated bone marrow biopsies of MN patients who underwent progression and compared them to a cohort with stable disease course. We identified certain mutations that promote an unfavorable outcome and found that patients with a known progress harbor more genetic alterations in their MN than those who do not deteriorate. Furthermore, we underpinned the hypothesis that not only the sum of genetic alterations but also the order in which they appear matters in disease evolution. Our findings emphasize the importance of genetic testing in MN patients in order to assess their risk of progression into aggressive blood cancer.

**Abstract:**

Myeloproliferative neoplasms (MPN) and myelodysplastic/myeloproliferative neoplasms (MDS/MPN) both harbor the potential to undergo myelodysplastic progression or acceleration and can transform into blast-phase MPN or MDS/MPN, a form of secondary acute myeloid leukemia (AML). Although the initiating transforming events are yet to be determined, current concepts suggest a stepwise acquisition of (additional) somatic mutations—apart from the initial driver mutations—that trigger disease evolution. In this study we molecularly analyzed paired bone marrow samples of MPN and MDS/MPN patients with known progression and compared them to a control cohort of patients with stable disease course. Cases with progression displayed from the very beginning a higher number of mutations compared to stable ones, of which mutations in five (*ASXL1*, *DNMT3A*, *NRAS*, *SRSF2* and *TP53*) strongly correlated with progression and/or transformation, even if only one of these genes was mutated, and this particularly applied to MPN. *TET2* mutations were found to have a higher allelic frequency than the putative driver mutation in three progressing cases (“*TET2*-first”), whereas two stable cases displayed a *TET2*-positive subclone (“*TET2*-second”), supporting the hypothesis that not only the sum of mutations but also their order of appearance matters in the course of disease. Our data emphasize the importance of genetic testing in MPN and MDS/MPN patients in terms of risk stratification and identification of imminent disease progression.

## 1. Introduction

The term myeloproliferative neoplasm (MPN) subsumes a group of hematopoietic stem cell disorders that are all characterized by clonal expansion of one or more myeloid lineages. With the discovery of distinct driver mutations in MPN, molecular analyses have gained immense importance in terms of diagnosis, follow-up and prognosis [1]. In *BCR*-*ABL1*-negative MPN, one of the canonical driver mutations, i.e., *JAK2*, *CALR*, *MPL* or—in cases of CNL—*CSF3R*, can be found in more than 90% of cases, these mutations mostly being mutually exclusive [2,3]. All these mutations directly or indirectly activate the JAK/STAT signaling pathway of the neoplastic clone [3,4]. Regardless of these known drivers, the acquisition of additional somatic mutations in MPN contributes to altered gene expression and is associated with a poorer prognosis, making the number of driver-independent somatic mutations one of the strongest predictors of outcome [1,3,5]. Moreover, not only the sum of mutations, but also the sequence, in which they are gained (i.e., before or after the actual driver mutation), influences the phenotype, subclonal plasticity and drug sensitivity and therefore might be of relevance for the individual patient [6,7,8]. The most frequently observed “non-driver” mutations affect epigenetic regulators such as *ASXL1*, *DNMT3A*, *EZH2*, *IDH1/2*, and *TET2*, leading to distinct effects on the transcriptional output, as might be expected from their physiological role [3,5]. Importantly, CHIP-associated mutations (*ASXL1*, *DNMT3A*, *TET2*) found at presentation do not seem to be significantly associated with transformation. By contrast, mutations of *IDH1/2*, *SRSF2* and/or *U2AF1* at first presentation are linked to progression [9].

A second group of myeloid neoplasms not only shows myeloproliferative features at the time of initial diagnosis, but also myelodysplastic characteristics, and is therefore designated as myelodysplastic/myeloproliferative neoplasm (MDS/MPN) [10]. In general, MDS/MPN frequently harbor mutations resulting in hyperactivation of the RAS/MAPK pathway (primarily *CBL*, *KRAS*, *NRAS* or *PTPN11*) [11,12,13], as well as mutations in epigenetic modifiers (most commonly *ASXL1* or *TET2*) [14,15,16,17] and splicing factors (*SRSF2* mutations being present in nearly half of chronic myelomonocytic leukemia (CMML) cases) [18], often combined with a MPN-characteristic driver mutation, as mentioned above.

In terms of disease evolution, every MPN or MDS/MPN has the potential for progression into ineffective hematopoiesis. This happens either by gain or enhancement of dysplastic features, by outcompetition of the normal hematopoiesis by the defective clone, or by bone marrow failure due to severe myelofibrosis (MF) [10]. Apart from that, one of the major sources of mortality in affected individuals is transformation into blast-phase MPN or MDS/MPN, a form of secondary acute myeloid leukemia (AML) [19]. Although the actual trigger events that lead to progression and/or transformation are yet to be determined, current concepts and single detailed case observations (e.g., [20]) suggest a branching stepwise acquisition of additional somatic mutations apart from—and occasionally independent of—the (initial) driver mutation that may result in or, at least, accompany disease evolution [21,22]. Mutations of *ASXL1*, *IDH1/2*, *RUNX1*, *SRSF2*, *TET2*, *TP53* and the *RAS* family genes, are found in AML secondary to MPN and in MPN gaining MDS-features, e.g., *TET2* and *SRSF2* being linked with emergent monocytosis in MPN, and *SF3B1* with emergent ring sideroblasts [17,20,23,24,25,26,27,28,29,30,31].

Genetic abnormalities typically found in de novo AML, e.g., *FLT3*-ITD or *NPM1*, are largely absent in blast-phase MPN [25]. Furthermore, and most interestingly, there is evidence of the development of driver—mainly *JAK2*—mutation-negative AML in driver mutation-positive MPN. This phenomenon can be explained by three possible scenarios: firstly, the evolution of a synchronous leukemic clone, independent from the actual MPN; secondly, the loss of *JAK2* mutation in the leukemic clone, as it might provide a differentiation signal hindering transformation; or thirdly, the presence of a common, clinically inapparent, pre-*JAK2* clone, e.g., with common clonal hematopoiesis of indeterminate potential (CHIP)-type mutations to give rise to both MPN and AML [32,33].

Analogously, distinct mutational patterns correlate with clinical outcome in patients with MDS/MPN. For instance, the presence of more than one alteration in the RAS signaling pathway is associated with an inferior event-free survival in juvenile myelomonocytic leukemia (JMML) compared to patients with only one RAS pathway mutation [34]. In atypical chronic myeloid leukemia (aCML), the presence of *TET2* mutations, as well as alterations in *SETBP1* seem to have an adverse impact on survival [35,36]. A correlation between the presence of mutated *TP53* and adverse prognosis in myelodysplastic/myeloproliferative neoplasms, unclassifiable (MDS/MPN-U) has been observed as well. In CMML, a high mutational burden, more than 3 mutated epigenetic regulators and the presence of *NRAS* mutations are associated with disease relapse [37], whereas survival is adversely affected by the presence of *ASXL1* mutations [38]. Finally, in *SRSF2* P95 mutant myeloid neoplasms mutations of *STAG2*, *RUNX1* or *IDH1/2* are associated with blast phenotype [31].

Although much has been achieved in the last decade in terms of deciphering the underlying molecular mechanism regarding the clinical and morphological heterogeneity of MPN and MDS/MPN, by far not all variables that influence progression and transformation have been identified. In this study, we investigated 13 MPN as well as 7 MDS/MPN patients with known progression/transformation and compared them to a control cohort of 11 patients with stable diseases. 15 paired samples were available and comparative mutational analysis was performed in 11 matched pairs in order to add another piece of evidence to the molecular puzzle of progression in myeloid neoplasms.

## 2. Materials and Methods

### 2.1. Patients and Outcome

Bone marrow biopsies from 31 patients with MPN or MDS/MPN were obtained, from the archives of the Institute of Medical Genetics and Pathology at the University Hospital Basel, Switzerland, the Institute of Pathology of the University of Bern, Switzerland and from the Department of Pathology and Neuropathology of the University Hospital Tübingen, Germany.

The bone marrow trephine biopsies were formalin-fixed, paraffin-embedded and decalcified with ethylenediaminetetraacetic acid according to international guidelines [39]. Hematoxylin-and-eosin- (H&E) and Gömöri-stained slides were prepared of each specimen and the degree of MF was assessed [40]. Immunohistochemistry was performed, where necessary, to establish the diagnosis (Figure 1). All cases were reviewed by two of the authors (BF and FF) to confirm the diagnosis and were classified according to the current WHO classification [10]. The study was approved by the Ethics Committee of the University Hospital Tübingen (106/2013BO2).

From the 31 patients, 20 progressed during the follow-up, whereas 11 had a stable disease course. Disease progression was defined as either gain (or enhancement) of MDS-like features, and/or the transformation into AML or myelosarcoma. In 16 instances, the biopsy from the time point of initial diagnosis was available. In 10 cases, the first biopsy was taken at later point of time, without information on eventually applied therapies (from one patient the biopsy was not available anymore as this was a consultation case and the material was sent back (case 13)). In the remaining 5 cases, there was no data whether the biopsy was taken at the initial time point or during the course of disease. From 15 patients, biopsies from two different time points during disease course were available. For further details, we refer to Table 1, displaying all relevant patient data as well as the exact diagnosis.

### 2.2. Molecular Analysis of Bone Marrow Biopsies

DNA was extracted from the bone marrow biopsies. 5 μm paraffin sections were generated, dewaxed and digested by proteinase K using the Maxwell RSC FFPE Plus DNA Kits (AS1720) and the Maxwell RSC tool (Promega, Mannheim, Germany) according to the manufacturer’s instructions.

Targeted mutation analysis was performed by Next Generation Sequencing (Ion GeneStudio S5 prime, Torrent Suite Software Version 5.10, Thermo Fisher Scientific, Waltham, MA, USA) using an AmpliSeq Custom Panel (Ion AmpliSeq Designer v7.4.10, Thermo Fisher Scientific, Waltham, MA, USA) (hotspot regions in 21 genes: *ASXL1*, *BRAF*, *CBL*, *CSF3R*, *CALR*, *ETNK1*, *FLT3*, *GNAS*, *HRAS*, *IDH1*, *IDH2*, *JAK2*, *KIT*, *KRAS*, *MPL*, *NRAS*, *SETBP1*, *SF3B1*, *SRSF2*, *STAT3*, *U2AF1*, complete coding sequence of 11 genes: *ATRX*, *CEBPA*, *DNMT3A*, *EZH2*, *IKZF1*, *NPM1*, *RAD21*, *RUNX1*, *TET2*, *TP53*, *ZRSR2*) or the commercially available Oncomine™ Myeloid Research Assay (Thermo Fisher Scientific, Waltham, MA, USA). Amplicon library preparation and semiconductor sequencing was carried out according to the manufacturers’ manuals using the Ion AmpliSeq Kit for Chef DL8, the Ion 510, Ion 520 and Ion 530 Kit—Chef and the Ion 530 Chip Kit on the Ion Chef and the Ion GeneStudio S5 Prime system (Thermo Fisher Scientific). Output files were generated with Torrent Suite 5.12.0. Variant calling of non-synonymous somatic variants compared to the human reference sequence was performed using Ion Reporter Software (Thermo Fisher Scientific, Version 5.16.0.2). Variants called by the Ion Reporter Software were visualized using the Integrative Genomics Viewer (IGV; Broad Institute, Cambridge, MA, USA; Version 2.8.0) to exclude panel-specific artefacts.

Interpretable results were achieved in 42 samples. Mutations found by either the AmpliSeq Custom Panel or the Oncomine™ Myeloid Research Assay were reported. Only mutations that were evaluated as “pathogenic” or “likely pathogenic” by VarSome [41] were taken into account in the (statistical) analysis.

### 2.3. Statistical Analysis

All statistical work-up including descriptive statistics was performed with the SPSS 25.0 software package (Armonk, NY, USA). Categorical data (i.e., presence of mutations) were summarized using frequency counts and percentages. Differences were tested with the paired sample *t*-test, Fisher’s exact test or Mann-Whitney-U-test, as appropriate. Spearman’s rank order correlation was used for calculation of correlation coefficients. For progression-free survival analysis, the types and numbers of events were summarized descriptively, and the prognostic role of mutations was calculated using the Kaplan-Meier method. Survival analysis was feasible in 21 out of the 31 patients. *p*-values below 0.05 were considered significant, such below 0.1—as trend; whenever possible, 2—sided test were applied.

## 3. Results

### 3.1. Patients and Outcome

20 patients underwent disease progression, of whom 13 were initially diagnosed with MPN and the remaining with MDS/MPN. Five of the MPN Type of Progression (Either Histologically an AML, two a myelosarcoma and one was diagnosed with 15% peripheral blasts, which was rated as progression (case 17). The control cohort (defined as MPN or MDS/MPN without progression) encompassed 10 MPN patients and one MDS/MPN case. For further details, we refer to Table 1.

For 12 patients with MPN and documented disease progression, the time between the initial diagnosis and transformation/progression was available; the median interval being 47 months (mean 67 months, range 2–167). Patients with a MDS/MPN did so within a median interval of 47 months (mean 47 months, range 23–70). For the seven patients without progression and known follow-up period the median observational duration was 85 months (mean 75 months, range 10–115).

### 3.2. Outcome and Total Number of Mutations

Molecular analysis of the 42 samples of the 31 patients yielded interpretable results with at least one of the applied panels. In 38 samples of 28 patients, at least one detectable pathogenic mutation was found. For MPN, in 28 samples of 23 patients, at least one detectable pathogenic mutation was found. All details regarding the specific type of mutation and their allelic frequencies are summarized in Figure 2.

Cases with progression/transformation, regardless of initial diagnosis, displayed a trend towards higher number of mutations at first biopsy (median 2 (range 0–4)) compared to those remaining stable (median 1 (range 0–3); *p* = 0.093, Mann-Whitney-U-test); in 3 cases of the control cohort and in one case of the progressed cohort, no mutations were found (cases 26; 29; 31 and case 14, respectively). Regarding MPN only, cases that run a stable disease course displayed at first biopsy 0–3 mutations (median 1, mean 1.3), compared to 1–4 (median 1, mean 1.85) in instances that later-on progressed, but this was not statistically significant. The second biopsy of progressed cases showed an increase in the total number of mutations compared to the initial total amount of mutations (from median 1 (0–3) to median 2.5 (1–7); *p* = 0.043, paired sample *t*-test). This roughly applied to MPN cases as well (from median 1 to median 1.5, *p* = 0.066). Considering the type of progression, cases that transformed directly into an AML displayed a higher number of initial mutations (median 3 (0–4)) compared to those progressing into MDS (median 1 (1–2)), and this reached statistical significance for MPN cases (median 2.5 (1–4) compared to median 1.5 (1–2), *p* = 0.045).

### 3.3. Correlation Analysis: Clinical Presentation and Presence of Specific Mutations

Among all investigated genetic alterations, we identified mutations in five different genes (*ASXL1* present in 3/19 progressed/transformed cases, *DNMT3A* in 2/19, *NRAS* in 2/19, *SRSF2* in 4/19 and *TP53* in 4/19, and being cumulatively present in 10/19 progressed/transformed cases) that strongly correlated with disease progression or/and transformation into AML, even if only one of these genes was mutated, since these mutations were not observable in the 11 cases that remained stable (*p* = 0.003). Excepting *DNMT3A*, all these genes applied to MPN in particular. When these five genes were considered individually, only the presence of *TP53* (*p* = 0.071, Fisher’s exact test) and *SRSF2* (*p* = 0.071, Fisher’s exact test) mutations appeared to be potentially linked to disease progression, and *TP53* (*p* = 0.081) applied to MPN in particular.

Taking into consideration time to progression, cases exhibiting at least one of these prognostically unfavorable mutations displayed a median progression-free survival of 17 months (mean 38), which was shorter compared to progressing/transforming cases without mutations in the respective genes (median 39 months, mean 84; *p* = 0.078). When compared to all cases (i.e., progressed cases without respective mutations and cases running a stable course), the discrepancy became all the more apparent, as the median progression-free survival was 17 compared to 146 months (*p* = 0.003; Figure 3A), which particularly applied to MPN (*p* = 0.001; Figure 3B).

### 3.4. Allelic Frequencies and Clonal Evolution

An overview of the allelic frequencies of the identified mutations of all cases can be found in Figure 2. Admittedly, no sufficient cytogenetic data was available to address duplications and/or deletions of the respective loci. Here we will only refer to the most notable results.

Four cases of the control cohort were found to have a detectable mutation in more than one gene at the time point of the first biopsy. Interestingly, of these four cases two had a *TET2* mutation additionally to a driver mutation *(CALR* in case 24; *JAK2* in case 28), which—taking into account the allelic frequencies—most probably developed in a driver-positive subclone (“*TET2*-second”).

In contrast, three progressed cases exhibited *TET2* mutations with a higher allelic frequency than the actual driver mutation (*JAK2* in cases 10, 12 and 19), which makes the emergence of the *TET2* clone before the *JAK2* mutation likely (“*TET2*-first”). One additional MDS/MPN-U case in the progressed cohort (case 15) displayed four different somatic mutations (*DNMT3A*, *ETV6*, *SRSF2* and *TET2)* and was found to have the highest allelic frequency in the *TET2* clone, suggesting itself to be a “*TET2*-first” case as well.

Furthermore, two MDS/MPN cases that underwent progression (case 19 into AML/myelosarcoma; case 17 in form of increased peripheral blasts) developed mutations typically found in primary AML and rather unusual for AML secondary to MDS/MPN (*NPM1* in case 19; *FLT3*-ITD in case 17). Notably, the two mutations found in the first biopsy of case 17 (*ASXL1* and *RUNX1*) were again detectable in the biopsy that followed, together with additional genetic alterations in *ETV6*, *SRSF2*, and the above mentioned *FLT3*, suggestive of linear disease progression. In contrast, case 19 exhibited three different mutations at the time point of the first biopsy (*CBL*, *JAK2*, *TET2*), whereas only two of them (*CBL* and *TET2*) were retained in the biopsy two years later, together with additional mutations in *NPM1*, *PTPN11* and *SRSF2*, suggesting that the *JAK2*-positive subclone got lost during branching disease progression.

## 4. Discussion

Although many enigmas regarding the molecular basis and development of MPN and MDS/MPN have been solved in recent years, and some genetic risk factors for progression have been identified, the mechanisms leading to disease progression still remain elusive. In this study we investigated paired biopsy samples of MPN and MDS/MPN, sequentially obtained over time, concerning their molecular alterations and compared them to a stable, non-progressive control cohort. Though this was a retrospective and targeted approach with only a limited number of cases, we confirmed observations made by others and extended information that could eventually contribute to our understanding of disease progression in MPN and MDS/MPN.

Even in our small collective, the co-existence of different mutations (cumulation of mutations) appeared to be prognostically unfavorable, as described elsewhere (e.g., [1] and [5]). We could show that progressed patients from our collective had an increased number of mutations already in the first biopsy compared to the control collective. In contrast to the observations made by Lundberg et al. [5], there was a significant increase in the number of mutations in the second biopsy of progressed patients compared to the first sample. This might be explained by the fact that the focus of our study was on patients with progression, who might have a higher propensity for genetic instability per se than stable patients, and the latter subgroup was rather small. Still, our observations are in concordance with a study performed by Senín et al. [42], showing that patients with progression have a significantly higher mutation rate than stable ones and that patients with additional mutations at the time of the first biopsy do have a higher risk of AML development.

In our cohort, we identified a set of distinct prognostically unfavorable mutations *(ASXL1*, *DNMT3A*, *NRAS*, *SRFS2* and/or *TP53*), that correlated with progression and transformation into AML. The prognostic impact of these mutations was particularly pronounced in MPN: in instances bearing such mutations, progressions occurred within a median of 1.5 years, while MPN cases lacking such mutations not only had lower frequency of transformation but also a median progression-free survival of over 10 years. Again, this partially reflects the insights already gained from other studies, as particularly *ASXL1*, *SRSF2* and *TP53* are known—among others—to be associated with a higher probability to develop AML in the background of MPN [5,9,42]. Other mutations described to be harbingers of adverse prognosis, such as *IDH1/2* or *RUNX1*, did not correlate with AML transformation in our cohort, which is most probably due to our limited sample size [42,43]. Additionally, our data suggest that *NRAS* mutations seem to have an adverse effect on clinical outcome. Fittingly, the presence of *NRAS* mutations is reportedly associated with adverse outcomes (disease relapses and/or shorter overall survival) in MPN as well as MDS/MPN [37,44,45,46,47].

It has been suggested by several groups that not only the number of mutations and types of mutant genes determine disease type and disease progression in myeloid neoplasms, but also the order in which they appear [7,8,33]. For example, Ortmann et al. described a predictive effect of *TET2* mutations to treatment with ruxolitinib in MPN patients, depending on the sequence of their appearance compared to the driver *JAK2.* For patients acquiring the *JAK2* mutation before *TET2* (“*TET2*-second”), a higher probability of a favorable ruxolitinib response compared to “*TET2*-first” patients was noticed. Furthermore, the authors suggested that a background *TET2* mutation alters the transcriptional effects of *JAK2* and therefore prevents the upregulation of its proliferative program, potentially resulting in a weaker “MPN-phenotype”, which may on its turn explain the decreased sensitivity towards ruxolitinib. In our cohort, we identified two “*TET2*-second” patients in the stable cohort with a MPN-phenotype (case 24 and case 28) and four “*TET2*-first” patients in the progressive cohort with a CMML (case 19), PMF (case 10 and case 12) and MDS/MPN-U (case 15) phenotype. Unfortunately, we cannot state on the patients’ sensitivity to ruxolitinib, as this drug was not yet approved by that time.

The two cases (case 19 and case 17) exhibiting AML-defining mutations during their clonal evolution represent good examples of the different proposed evolutional patterns along the transformation of a myeloid neoplasm into secondary AML: case 17 most probably linearly developing out of a clone carrying *ASXL1* and *RUNX1* mutation; in contrast, the *JAK2* mutant clone of case 19 most probably getting lost, potentially due to expansion and overgrowth of a *CBL*-mutant clone that additionally acquired a strong—de novo AML-type—driver mutation in *NPM1*, accompanied by *PTPN11* and *SRSF2* mutations equipping the respective clone with growth advantage. In addition of being illustrative of the evolutionary pattern of secondary AML, these two latter examples impressively show that, though rarely, de novo AML-typic driver mutations may occur in secondary AML and do not preclude evolution from MDS/MPN.

## 5. Conclusions

In summary, our data emphasize that genetic testing in MPN and MDS/MPN patients is of great importance for risk stratification and add some evidence that this may be important for treatment strategy planning, e.g., regarding ruxolitinib. Furthermore, repetitive testing during the course of the disease might help to predict imminent disease progression if the acquisition of certain sets of additional mutations is observed or might help to explain a clinical disease progression that may not be simply reflected by numeric increase of blasts. In addition, this study documents the suitability of archival bone marrow specimens for the retrospective and sequential analysis of hematopoietic neoplasms and their evolution. This is especially useful in the analysis of MPN, which are frequently characterized by a very long disease course.

## Figures and Tables

**Figure 1 cancers-13-05605-f001:**
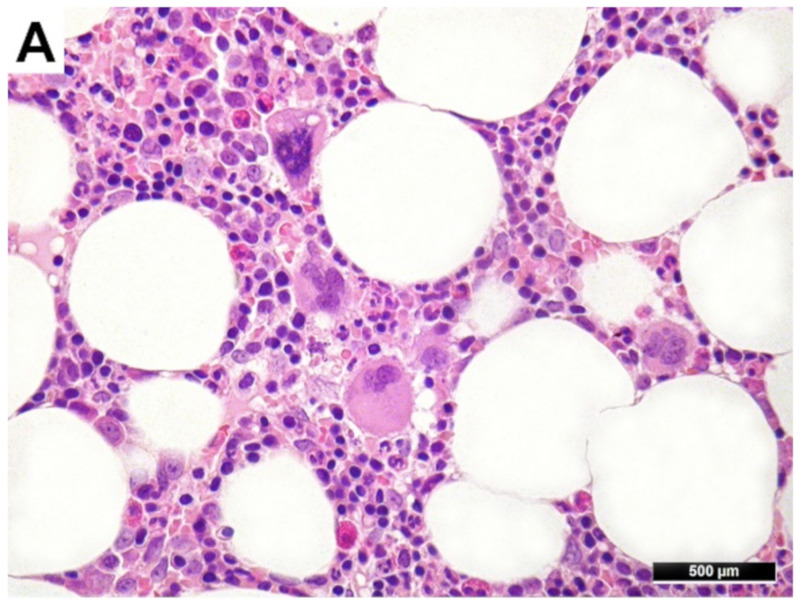
Sequential bone marrow of a myeloproliferative neoplasm) patient suffering from essential thrombocythemia (**A**), who progressed to secondary acute myeloid leukemia (**B**) with increased CD34+ blasts (**C**) and clearly perceptible myelodysplastic features (micromegakaryocytes, megakaryocytes with alobated nuclei and such with nuclear separations, CD34+ megakaryocytes).

**Figure 2 cancers-13-05605-f002:**
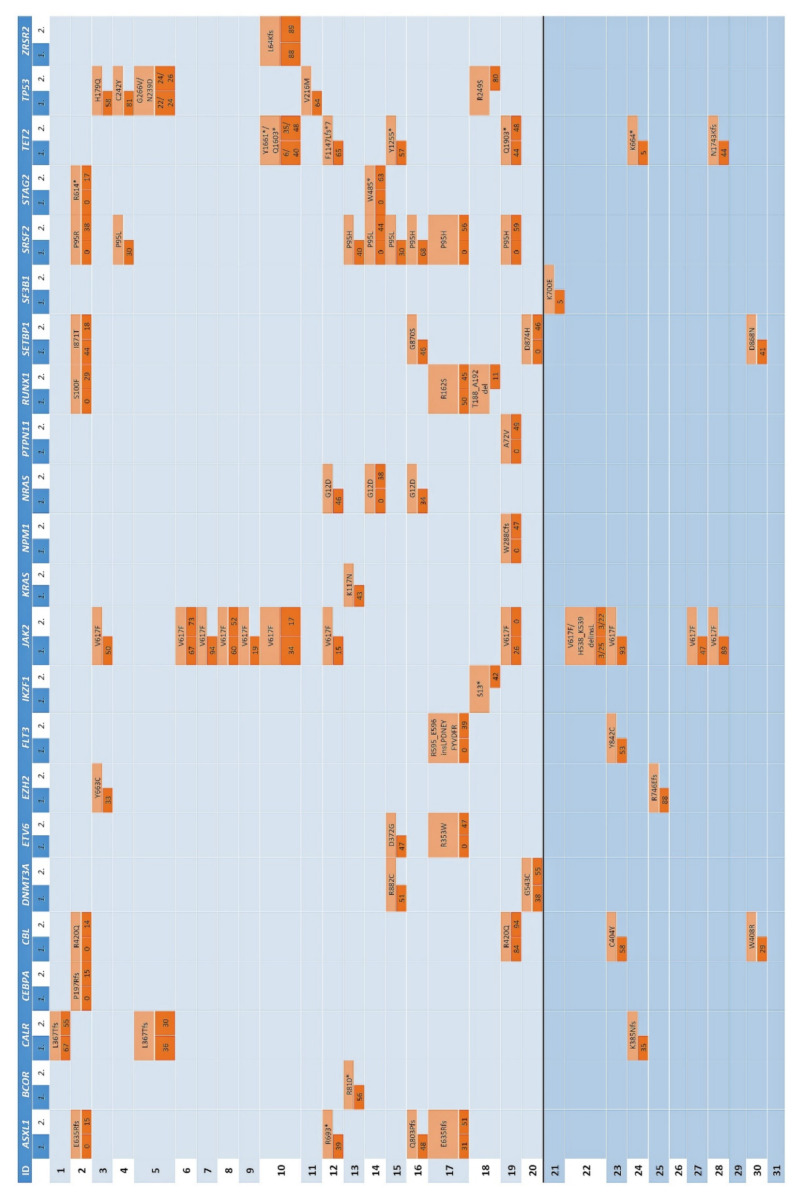
Results of the targeted mutation analysis performed either by an AmpliSeq Custom Panel, the commercially available Oncomine™ Myeloid Research Assay (Thermo Fisher Scientific), or by both. Only mutations rated as “pathogenic” or “likely pathogenic” by VarSome were considered. Each patient is represented by one line with an individual ID. Each column represents one gene that was mutated in at least one sample. If mutated, the box is highlighted in orange and the according human genome variant society-nomenclature on protein level is given in the upper part of the box. In the lower part, the respective allelic frequency of the first available biopsy is given on the left side (subcolumn “*1.*”), and the allelic frequency of the follow-up sample is shown on the right side (subcolumn “*2.*”). If a mutation was detected with both panels, the mean of the indicated allelic frequencies is reported. Boxes highlighted with orange and an allelic frequency of “0” represent cases without detectable mutation in the respective sample. If allelic frequency of a mutated gene is not given, molecular work-up on the respective sample was not possible (blue background).

**Figure 3 cancers-13-05605-f003:**
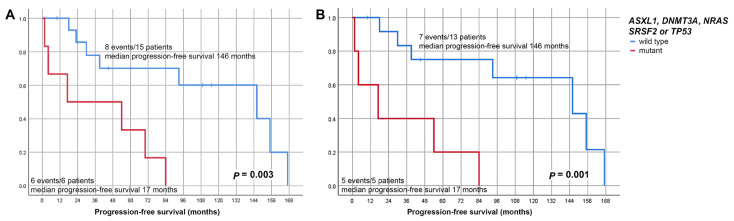
Progression-free survival of all MPN and MDS/MPN cases (**A**) and MPN cases only (**B**), bearing mutations in either *AXSXL1*, *DNMT3A*, *NRAS*, *SRSF2* or *TP53* compared to that of cases that did either not progress (these cases did not bear such mutations) or progressed but did not display such mutations.

**Table 1 cancers-13-05605-t001:** Patient characteristics. Abbreviations: aCML, atypical chronic myeloid leukemia; AML, acute myeloid leukemia; CMML, chronic myelomonocytic leukemia; ET, essential thrombocythemia; F-up, follow-up; MDS, myelodysplastic syndrome; MPN, myeloproliferative neoplasm; PMF, primary myelofibrosis; PV, polycythemia vera; U, unclassified. “Sample 1” describes the first available biopsy, which was either gained at the time point of “Initial diagnosis” or later in disease course (“F-up”). It was used to assess baseline molecular characteristics or, in cases with only an available biopsy at progression, to assess the mutational landscape of the respective progressed disease. Cases from which we did not know, whether it was a biopsy from the initial diagnosis or a follow-up, are designated with “?”. All histological diagnoses are shown in brackets. “Sample 2” describes the second follow-up biopsy (if available) with the according histological diagnosis. Boxes with a grey background display biopsies that yielded interpretable molecular results.

ID	Sex	Age	Primary Disease	Progression	Type of Progression (Either Histologically Confirmed or Clinically Documented)	Time to Progression or Last Follow-Up (Months)	Sample 1	Sample 2
1	F	57	ET	Yes	MDS > AML	146	Initial diagnosis	F-up (MDS)
2	F	67	ET	Yes	MDS > AML	39	Initial diagnosis	F-up (AML)
3	F	71	ET	Yes	AML	unknown	F-up (MPN blast phase)	Not available
4	M	63	ET	Yes	MDS	54	F-up (MDS EB1)	Not available
5	M	55	ET	Yes	AML	84	F-up (MPN blast phase)	F-up (AML)
6	M	49	PV	Yes	MDS > AML > myelosarcoma	155	F-up (MDS)	F-up (Myelofibrosis)
7	M	70	PV	Yes	MDS	93	F-up (MDS)	Not available
8	F	53	PMF	Yes	AML	167	Initial diagnosis	F-up (PMF)
9	F	58	PMF	Yes	AML	30	Initial diagnosis	Not available
10	M	74	PMF	Yes	AML	18	Initial diagnosis	F-up (AML)
11	M	55	PMF	Yes	AML	4	Initial diagnosis	Not available
12	M	71	PMF	Yes	AML	1.5	F-up (accelerated MPN)	F-up (AML)
13	M	55	MPN-U	Yes	AML	17	Not available	F-up (AML)
14	M	71	MDS/MPN-U	Yes	AML	23	Initial diagnosis	F-up (AML)
15	M	80	MDS/MPN-U	Yes	AML	unknown	? (MDS/MPN-U)	Not available
16	M	83	MDS/MPN-U	Yes	AML	unknown	? (MDS/MPN-U)	Not available
17	M	51	MDS/MPN-U	Yes	MDS excess blasts (EB) type 2	70	F-up (MDS/MPN-U)	F-up (MDS/MPN-U)
18	M	55	CMML	Yes	AML > myelosarcoma	unknown	? (CMML)	F-up (CMML)
19	F	79	CMML	Yes	AML > myelosarcoma	unknown	? (CMML)	F-up (CMML)
20	M	78	aCML	Yes	AML	unknown	? (aCML)	F-up (aCML)
21	M	76	ET	No	None	unknown	Initial diagnosis	Not available
22	M	45	PV	No	None	109	F-up (PV)	F-up (PV)
23	M	75	PV	No	None	109	Initial diagnosis	Not available
24	F	64	PMF	No	None	115	Initial diagnosis	Not available
25	F	84	PMF	No	None	60	Initial diagnosis	F-up (PMF)
26	F	82	PMF	No	None	unknown	Initial diagnosis	Not available
27	F	75	PMF	No	None	45	Initial diagnosis	Not available
28	M	67	MPN-U	No	None	unknown	Initial diagnosis	Not available
29	M	30	MPN-U	No	None	unknown	Initial diagnosis	Not available
30	M	75	MPN-U	No	None	10	Initial diagnosis	Not available
31	M	68	MDS/MPN-U	No	None	24	F-up (MDS/MPN-U)	F-up (MDS/MPN-U)

## Data Availability

Data may be obtained from the corresponding author on request.

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
