# Peer review of "Molecular Progression of Myeloproliferative and Myelodysplastic/Myeloproliferative Neoplasms: A Study on Sequential Bone Marrow Biopsies"

_cancers, 2021, doi:10.3390/cancers13225605_

Round 1

Reviewer 1 Report

The authors review current classification of the MPNs and MDS/MPN overlap syndromes in the introduction, I think this can be shortened. Instead, I would focus on recent papers describing clonal evolution and order of acquisition of mutations and less on the classification. Many other manuscripts have been published delineating progression from MPN to advanced disease:  Lundberg et al Blood 2014, Chang Blood 2014, Engle Leukemia 2015, Tefferi et al published on TET2 mutations in MDS/MPN in 2009 in Leukemia. It is difficult to conclude much from such a heterogeneous group of patients. I am not sure why both MPN and MDS/MPN patients were grouped together. I would favor sticking to one group of patients only or analyzing the patients separately. Determining what makes an ET progress to AML is unlikely the same as what makes an MDS/MPN patient progress. In addition, the authors make some conclusions based on very little information- for example, one can't conclude that JAK2 mutated patients have a higher rate of myeloid sarcoma progression because 2 patients with myeloid sarcoma had a JAK2 mutation. Much larger numbers are needed to conclude this. Lastly, it wasn't clear why this group of patients was chosen in the first place. 

The authors have paired bone marrow samples on MPN and MDS/MPN overlap patients as well as information on NGS testing on the paired samples, which is very interesting.

The description in page 5 and section 3 of samples tested is difficult to follow. Particularly in section 3, it is hard to keep track by reading the paper how many patients are part of the transformation group (MPN and MDS/MPN) and how many are the stable controls.  

Reviewer 2 Report

Brune and colleagues present a study focused on the characterization of the molecular features of progressed MPN and MDS/MPN. The main limitation of this study is the small number of patients included, although this is partially compensated by the fact that some of them are analyzed in sequential samples, which is always interesting. Overall, the study is well designed and conducted, although there are several aspects that need to be clarified in the manuscript.

  • The introduction is very well-written and provides sufficient background. Minor comments:
    1. When describing the correlation of mutations and clinical outcome in CMML (lines 121-122), I think that the negative impact of ASXL1 should be mentioned, since it is the only gene that has been consistently associated with unfavorable outcomes in this disease (and thus is included in current CMML-specific prognostic scores).
    2. Lines 130-131: why comparative analysis was performed in 11 matched pairs if 15 paired samples were available?

  • The methods are adequately described, but the following should be clarified:
    1. Line 150: there are 14 patients without a biopsy from the initial diagnosis but a known history of MPN (n=7) or MDS/MPN (n=7) and one available follow-up biopsy. I understand that this follow-up biopsy was used to analyze the baseline molecular profile of these patients. Were these biopsies prior to any treatment? Otherwise, the results should be carefully interpreted, or interpreted accordingly.
    2. Molecular analysis: the authors write “interpretable results were achieved in 42 samples”. What exactly do they mean by interpretable? Of the total of samples analyzed, how many did not have interpretable results and why these results could not be interpreted?
    3. Table 1: I understand that patients with “unknown” time to progression or to last follow-up could not be included in the survival analysis (Kaplan-Meier). This could be specified in the manuscript (either in the methods or the results). For example, in the “statistical analysis” paragraph (Lines 196-199) the following could be added: “survival analysis was performed in XX/XX patients with follow-up clinical data, using the Kaplan-Meier method…”
    4. Table 1: I would suggest to include in this table the information of the available samples, to facilitate the understanding of the results, which is sometimes confusing. For example, add two columns “Sample 1” and “Sample 2/Follow-up sample” and then specify, for each sample, the timepoint at which the samples were obtained, for example:
      1.  

      Sample 1

      Sample 2 (follow-up)

      Diagnosis

      Not available

      Follow-up (not treated)

      Progression to AML

      Diagnosis

      After treatment with…

  • Overall, the presentation of the results is correct but could be improved. For that, I would suggest the following modifications/clarifications: 
    1. I would suggest to include a small paragraph describing the global molecular profiling results, to facilitate the understanding of the results that will be presented later. For example, before section 3.2 or at the beginning of this section (Line 219), a short sentence could be included presenting the number of mutations detected, the mean number of mutations per patient…etc, and refer to the current Figure 3, where the details of the molecular profile of each patients can be seen (this would be then Figure 2).
    2. Lines 220-221: the authors state that “Cases with progression/transformation, regardless of initial diagnosis, displayed a trend towards higher number of mutations”. This analysis was performed using the data of those patients with available sample at diagnosis or also those patients without a sample at diagnosis (where a follow-up sample was used)? In that case, using samples that might be after treatment or without treatment but years after diagnosis might not be accurate to perform that type of comparison.
    3. Lines 237-240: it should be specified or better clarified that the prognostic impact of these genes is not individual, and that the five genes are considered cumulatively. Otherwise it can be confusing, since it is previously mentioned that only the presence of TP53 and SRSF2 mutations appear to be linked to progression.
    4. Regarding the clonal evolution analysis, that was performed by comparing variant allele frequencies (VAFs), did the authors used adjusted VAFs? VAFs need to be adjusted by copy number and zygosity if you want to use them to assess the size of the clone, compare with other mutations and establish the mutation order. For example, TET2 mutations often co-exist with chromosome 4q loss of heterozygosity (and less frequently TET2/4q microdeletion). In that case, TET2 real VAF would be half the VAF value that is obtained by NGS, which should be taken into account for comparing with another mutation in order to inferring the order of acquisition.
    5. Line 264: case 15 is described, but I would eliminate “without classical driver genes” from the sentence, since it doesn’t really apply for this case. According to Table 1, case 15 is an MDS/MPN and therefore a classical driver gene (JAK2, CALR, MPL) is not expected to be detected. Maybe it could be specified “one additional MDS/MPN case in the progressed cohort (case 15) displayed…”
    6. Figure 3: there is a typo, the legend says “Figure 2” and t corresponds to “Figure 3” according to the text. However, as mentioned earlier, I would suggest to move this figure to actual Figure 2, and present it before the survival analysis. Also, the quality of this figure needs to be improved (it is difficult to read it).
  • The discussion is well-written but the authors should be careful with some statement/conclusions that might be too strong considering the results here presented and the number of samples analyzed:
    1. Lines 295-296: the authors say that they compare the results of the progressed cases to “a stable, non-progressive control cohort”. However, this non-progressive cohort includes 11 patients, 4 of which have an “unknown” time to progression (therefore can’t be considered non-progressive) and one has a short follow-up (10 months) to be considered a stable non-progressive case. This results in a small cohort of 6 non-progressive patients. The “n” of this control group should be increased in order to do proper comparisons.
    2. Line 298: the authors state that they add “novel information”. Which is the novel information? In Lines 320-321 they state that, as a novel finding, the data suggests that NRAS mutations seem to have an adverse impact on clinical outcome. However, this is not exactly novel since the negative impact of NRAS has been previously described not only in CMML, which they mention, but also in PMF (Tenedini et al, Leukemia 2014; Santos et al, Leukemia 2020), ET and PV (Luque Paz et al, Blood Adv 2020) and aCML (Palomo et al, Blood 2020).
    3. Lines 333-337: the authors compare their results to the results published by Ortmann et al, but this should be done carefully, since that paper includes only MPN cases, and here 2/4 patients that are “TET2-first” are MDS/MPN, and it is known that clonal evolution of MDS/MPN is different from that of MPN. For example, MDS/MPN don’t have the same driver mutations as MPN (JAK2, CARL and MPL; in fact CALR and MPL are extremely rare), and moreover, in these patients mutations affecting myeloproliferative genes (ex. JAK2) are commonly secondary to mutations in epigenetic regulators (ex. TET2).

Round 2

Reviewer 1 Report

Although I appreciate the changes made to the article from its initial submission, I still contend that the sample size is too small to make any definitive conclusions. Although it is true that MDS/MPN and MPN patients share mutations and pathophysiology, this doesn't justify studying these patients together to understand progression. 

Author Response

We highly appreciate the input of the reviewer and would like to thank her/him for the criticism. 

Therefore, we performed subgroup analyses in the revised version of the paper, as suggested, and added the respective results. Furthermore, we provide a revised Figure 3 that illustrates the findings in MPN.

Reviewer 2 Report

Most of the previous comments/concerns were adequately addressed.

The quality of Figure 1 need to be improved (it is difficult to read).

Author Response

The problems concerning Figure 2, which we think the reviewer refers to, may be linked to pdf.-conversion. If displayed by an imageviewer program (jpg.), the quality seems fine to us. We suggest that the technical editor may feed-back us on the final quality, which will be displayed in the print version.